**On the drought in the Balearic Islands during the hydrological year 2015-2016**
Climent Ramis, Romualdo Romero, Víctor Homar, Sergio Alonso, Agustí Jansà and Arnau
Amengual
Meteorology Group. Department of Physics. University of the Balearic Islands- 07122 Palma.
Spain.
**Abstract**
During the hydrological year 2015-16 (September to August) a severe drought affected the
Balearic Islands, with substantial consequences (alleviated partially by desalination plants) on
water availability for consumption from reservoirs and aquifers and also on the vegetation
cover. In particular, a plague of 'Xilella fastidiosa' reached a relatively alarming level for the case
of the almond and olive trees. The expansion of this infestation could be attributed to, or at least
favored by, the extreme drought. In this paper we analyze this anomalous episode in terms of
the corresponding water balance in comparison with the balance obtained from long-term
climatological data. It is shown that the drought was the result of a lack of winter precipitation,
the lowest in 43 years, which led to a shortage of water storage in the soil. In several
meteorological stations analyzed, evaporation was greater than precipitation during all the
months of the year. In terms of attribution, it is found that during the 2015-16 winter the
atmospheric circulation over the North Atlantic was largely westerly and intense, with high
values of the NAO index that were reflected in high pressures over the Iberian Peninsula and the
western Mediterranean.
Keywords: drought, Balearic Islands, water balance, Mediterranean Sea.
1.   **Introduction.**
The Balearic Islands are located in the central part of the Western Mediterranean basin (Fig. 1).
The archipelago presents a well-marked interannual variability in the annual precipitation as it
was shown by Homar et al. (2010). Within this interannual variability, a particularly severe
drought episode occurred during the hydrological year (September to August) 2015-2016.
Actually, the drought affected the eastern part of the Iberian Peninsula as it was reported by the
Spanish                    Meteorological                    Agency                    (AEMET,
http://www.aemet.es/es/serviciosclimaticos/vigilancia_clima/). However, we restrict this study
to the Balearic Islands, where the population of perennials suffered a remarkable mortality,
especially among almond, olive and other fruit trees. Mostly in the southern part of the
archipelago shrubs and other plants such as bushes and steppes also perished, especially young
individuals which have very shallow roots. In addition, a plague of 'Xylella fastidiosa' reinforced
after the summer of 2016, and this could be attributed to, or at least favored by, the drought
and further hydrological stress suffered by the almond and olive trees. Although it is difficult to
assess quantitatively the total losses resulting from the drought (these could have reached more
than 10 million euros in the livestock breeding owing to loss of up to 90% of the production of
forage, according to the "Diario de Mallorca" newspaper, 4 June 2017), different lines of funding
were issued by the regional government. Besides the impacts on the natural and agricultural
systems, the demand of water for personal and leisure consumption reached its historical
maximum during the summer of 2016 (36.5 x $10^6$ m$^3$ during August 2016 in Mallorca, according
to same newspaper), when the islands registered a record number of tourists to date (more than
10.9 million in Mallorca). All together left the reservoirs and aquifers of the islands at levels of
great concern, putting at serious risk the supply for the following months in case of drought
persistence without the help of the desalination plants. This severe drought can be framed in
the context of the observed increase in the frequency of droughts in the Mediterranean area
(Hoerling et al. 2012) and in particular in the Spanish eastern lands (Vicente-Serrano et al. 1994).
The lands surrounding the north, east, and west of the Mediterranean Sea have a climate that
is characterized by a mild and rainy winter and a warm and dry summer. According to the
classification of Köppen these are thus considered to have a Csa type climate (Peel et al. 2007).
This type takes the generic name of Mediterranean climate. The Köppen classification global
map is determined from gross climatic features; when analyzing the data at higher resolution,
noticeable differences are found, even between contiguous areas of reduced extent. The
Balearic Islands (Fig. 1), with a typical Mediterranean climate, is a specific example of a context
exhibiting notable climatic differences within a relatively small region. Given the size of the
islands (Mallorca, the largest, extends over 3640 km$^2$), among all the influencing factors we must
attribute to the orography the greatest part of observed climatic differences over the territory.
These contrasts are indeed quite accentuated in the archipelago. The four major islands of the
Balearics have similar patterns of mean monthly rainfall but the spatial distribution of annual
totals is quite heterogeneous. Menorca and Ibiza-Formentera show a remarkable spatial
uniformity, with mean annual values higher in Menorca than in Ibiza-Formentera (Guijarro 1986,
Jansà 2014, López et al. 2017). These wetter conditions are attributed to the higher latitude of
Menorca, being the island more frequently affected by the fronts linked to the low-pressure
disturbances that evolve through Central Europe and by the lows developed over the Genoa
Gulf. In Mallorca there is high spatial contrast in the mean annual distribution of precipitation.
Along the southern coasts, where the orography is practically absent, annual precipitation
values are of the order of 350 mm on average, while in the zones with the highest mountains
(Tramuntana range, heights up to 1500 m, see Fig. 1) in the northwest of the island, the average
annual rainfall reaches 1400 mm (Guijarro 1986). These large contrasts occur within a distance
of about 50 km. In fact, attending to the climatic characteristics of the south of Mallorca, it rather
conforms to BSk type from the classification of Köpen, that is, winters not excessively dry,
temperate, but with very dry and torrid summers. The northern and northeastern zones of
Mallorca receive precipitations of the same order as those of Menorca, once again clearly above
the accumulations of the southern region.
Another characteristic of the rainfall over the Balearic Islands is its marked seasonality. The
ombrothermic diagram for the Mallorca airport (Fig. 2; Jansà et al., 2016) shows the most
outstanding feature of the Mediterranean climate: the above mentioned scarcity of
precipitation during the summer as well as relatively high temperatures during this period of the
year; also that autumn and winter are mild and relatively wet. The fact of reaching the end of a
hot summer after two months with almost no precipitation, somehow characterizes the type of
vegetation present in the lowlands (pines, shrubs, bushes and steppes but also almond trees).
At the same time, the islands have an economy fundamentally dependent on tourism (in 2016,
Balearic airports received 36.8 million passengers, according to the official web pages of the
three airports) that is mainly concentrated in the summer months. The supply of drinking water
during this period depends critically on underground aquifers (and on the supplementary action
of desalination plants) since existing reservoirs in the rainiest mountainous area of Mallorca are
too small. After the long and extreme summer, the recovery of the aquifers is strongly
determined by the amount of rainfall received during the autumn and the following winter. The
flora will be subjected to greater or lesser hydric stress depending mainly on the behavior of
autumn rainfall. The occurrence of large water stress situations is not uncommon, given the high
interannual variability that characterizes annual precipitations in the Balearic Islands (Homar et
al., 2010). Extreme manifestations of such variability are not new; there are written references
about important droughts affecting the archipelago during the Middle Age (Barceló, 1991), as
well as many oral references to the hazardous drought occurred during 1912-1913 in Mallorca,
a time when the local economy was almost exclusively dependent on agriculture.
Given the strong water deficit imposed to the vegetation by the end of the summer and also the
natural cycle of the underground aquifers, it may be more suitable to analyze precipitation in
terms of the hydrological year (September to August). Additionally, in order to account for the
vegetation stress in more detail, it becomes more informative to calculate the annual water
balance in which precipitation and evaporation are presented together (considering for the
latter the potential evapotranspiration) and to compare it with the climatic water balance for
which the local vegetation has adapted.
This paper presents in section 2 the interannual variability of the precipitation regime in the
Balearic Islands, both from the standard and hydrological year perspectives, as well as the
climatic water balance of the region. Section 3 discusses the water balance for the hydrological
year 2015-16 in detail. In section 4 the circulation pattern of the exceptional context that led to
the severe drought of that year is analyzed and compared with the pattern of an illustrative wet
year. Finally, section 5 presents the main findings and conclusions of the study.

**2.  Precipitation variability and climatic water balance.**

Monthly precipitation values at Mallorca, Menorca and Ibiza airports from 1973 to 2016 (44
years) have been analyzed. These are the longest climatic series without gaps in the Balearic
Islands. From the monthly values, annual accumulations as well as those corresponding to the
43 hydrological years from 1973-74 to 2015-16 have been calculated.
The anomalies of the annual rainfall with respect to the average of the reference period 1981-
2010 for the airports of Mallorca, Menorca and Ibiza have been considered (not shown). The
yearly mean for the reference period at Mallorca is 411.3 mm and the interannual variability of
the series is large enough as to yield a standard deviation of 100.9 mm (coefficient of variation
CV=24.5%). The average for Menorca airport is 548.6 mm and the standard deviation is 132.8
mm (CV=24.2%). These values for Ibiza are 411.1 mm and 117.3 mm (CV=28.5%). These
relatively large values of the coefficients of variation reveal the high interannual variability of
precipitation in the islands, which is itself related to the variability of the atmospheric patterns,
as shown in Section 4.
As revealed by the CV values, the variability is greater in Ibiza than in Menorca, although there
are anomalies in both stations that occasionally exceed 200 mm. It is noteworthy the relatively
low correlation (0.54) that exists between the time series of Mallorca and Menorca, but
especially low is the correlation between the time series of Menorca and Ibiza (0.30). For
Mallorca and Menorca there are few cases in which a positive anomaly in one station does not
correspond to the same sign in the other. One of such cases is 2016, when the intense rainfall
recorded in Mallorca during the months of October and December (107.6 and 150.4 mm,
respectively) explains the positive anomaly of its airport; however, this event did not affect
Menorca (13.2 mm and 79.8 mm, respectively).
It should be noted that a wet/dry year at the airports tends to be accompanied by greater/lower
than normal annual precipitation in the rest of each respective island. Figure 3 shows this kind
of distribution for the years 1996 and 1999, considered as wet and dry years, respectively.
However some kind of objective index should be applied to analyze the representativeness of
the interannual variability of the rainfall captured by the airports, especially in Mallorca where
the spatial variability of the annual rainfall is very high as previously indicated (Guijarro 1986).
An analysis of the spatial representativeness of the interannual variability captured by the
Mallorca airport has been performed using two methodologies. First, the time series of the
relative annual anomalies (anomaly divided by the corresponding annual average) have been
calculated for five meteorological stations located in Mallorca, and the resulting mean time
series (of the five individual series) has been determined. The five stations are representative of
different pluviometric regimes of the island: mountainous area, north, center, east and south.
For this analysis, the period 1981-2010 has been considered. The time series of annual relative
anomalies at Mallorca airport has been compared against the above mean time series. The time
series exhibit a correlation coefficient as high as 0.9. The second method is analogous to the
previous one but uses the precipitation analyses across the island of Mallorca that were derived
in the PREGRIDBAL project (López et al. 2017). These analyses have a resolution of 100 m and
use all available observed data for each product requested. Annual precipitation grid data has
been considered for each of the years 1980-2009, together with the grid analysis of mean
precipitation corresponding to these 30 years. For each grid point and for each year the relative
annual anomalies have been determined and a time series expressing the spatial average of
annual anomalies has been calculated. Finally, this time series has been compared against the
relative anomalies at Mallorca airport, yielding in this case a correlation coefficient of 0.86 (Fig.
4). Thus, it seems well justified the assumption that the spatial-temporal variability in the island
of Mallorca is correctly captured by the series of precipitations at the Mallorca airport.
Due to their relatively small size and moderate orography, the spatial variabilities of the annual
mean precipitation in Menorca and Ibiza are much lower than in Mallorca, therefore it seems
clear that the corresponding time series at the airports are even more representative of the
corresponding interannual variability of the whole islands.
Figure 5 shows the precipitation anomalies at the airports of Mallorca, Menorca and Ibiza for
the hydrological years 1973-74 to 2015-16 (43 years) with respect to the reference period 1980-
81 to 2009-10. Recall the hydrological year comprises from September to August. The mean
precipitation for the reference period in Mallorca is 409.5 mm, with a standard deviation of
119.2 mm (CV=29.1%). For Menorca these values are 544.3 mm and 120.5 mm (CV=22.1%). For
Ibiza 413.0 mm and 116.6 mm (CV=28.2%) respectively. Mean values are very similar with
respect to the observations derived from the "standard" or natural years but the interannual
variability is higher now in Mallorca and lower in Menorca. In the present case there is a greater
correlation (0.68) between the anomalies of these two rainfall stations. In Ibiza the values are
very similar to those obtained for the natural year. The correlation between the time series of
Menorca and Ibiza is identically low (0.33 vs 0.30 for the natural years). These low correlations
values are a clear manifestation that the rain bearing meteorological systems for the north and
south of the archipelago do not respond to the same circulation patterns, as previously reported
by Guijarro (2002, 2003). A detailed study on the surface circulation related with daily rainfall
patterns in Mallorca can be found in Sumner et al (1995).
Looking at Fig. 5, it can be observed that dry hydrological years leading to water stress on the
flora, and probably on the aquifers, become clearly distinguishable. The periods 1981 to 1984,
1991 to 1994 and to 1998 to 2001 are noteworthy. It can be observed that in 2015-16 there are
also negative anomalies, much more important in Menorca.
Although there are several indices to characterize a drought (e.g. the Palmer Drought Severity
Index (PDSI), Palmer (1965); the Standardized Precipitation Index (SPI), McKee et al. (1993);
Supply Demand Index (SDDI), Rind et al. (1990)), from an ecological point of view and in order
to account for the possible water stress on the flora, it is interesting to analyze the water balance
directly, in which the precipitation is compared against the evaporation, month by month, and
from this balance to evaluate the periods of the year in which there is an excess or lack of water
in the soil. In this sense there are studies on the effects of droughts on the Mediterranean flora
in Spain (e.g. Peñuelas et al. 2001). The determination of the potential evapotranspiration (PET)
is an important step when estimating soil water deficit or excess. However, empirical formulas
for estimating PET have their limitations, the results cannot be considered at the same level of
exactitude as precipitation measurements. In consequence, the comparison between
precipitation and PET has to be regarded as an approximation to the reality. The existence of
several analytical expressions to calculate PET using different variables, also demonstrate the
difficulty to determine this magnitude accurately.
Estimation of the climatic water balances at the three airports was carried out using the
Thornthwaite method (1948) for the determination of monthly potential evapotranspiration,
using monthly mean temperature and precipitation values referred to the reference period
1981-2010. In our analysis, actual evaporation is considered to coincide with calculated potential
evapotranspiration if monthly precipitation is greater than potential evapotranspiration, and in
these circumstances the remaining precipitation is converted to water stored in the soil. These
amounts can be cumulative through the year and if the total storage reaches a value which is
considered to be the maximum capacity of the soil, the excess becomes surface runoff and
infiltration. The maximum storage of the soil depends on several factors, e.g. the texture, land
use and slope of the terrain. Botey and Moreno (2015) have produced a map of the soil
maximum storage for the Iberian Peninsula and the Balearic Islands. From the information
displayed in their map, for the low lands of the Balearic Islands, where the used meteorological
stations are located, 100 mm can be considered a reasonable value. If the monthly precipitation
is less than the potential evapotranspiration, the actual evaporation is equal to the precipitation
plus the reserve portion of the soil moisture that is needed, until it is exhausted. The remaining
difference between potential evapotranspiration and actual evaporation is indicative of the
water deficit that has to be overcome by vegetation. Balance calculations begin in the month of
September, considering that the soil does not contain any water after the dry summer.
Figure 6 shows the climatic water balance (1981-2010) during the hydrological year, according
to the indicated method, for the airports of Mallorca, Menorca and Ibiza. Climatologically, there
is deficit in Mallorca for the first month of September. There is storage of water in the soil from
October to February, which is totally exhausted by the end of June. During the summer (June-
August) the deficit is very large, reaching 150 mm. At the Menorca airport there is also a deficit
in September, the accumulation of water in the soil begins in October, and there is runoff or/and
infiltration during January, February and March. The water stored in the soil of Menorca allows
for evaporation to be larger than precipitation even in June, with a total lack of soil water
observed only in July and August. The maximum deficit also reaches 150 mm. At Ibiza the water
balance is very similar to Mallorca but the storage of water in the soil during the winter is lower
and therefore it is consumed more quickly, inducing a large deficit during all the summer.
The climatic water balance at Menorca and Ibiza airports can be considered representative of
the whole islands. In contrast, for the larger and more complex island of Mallorca it is evident,
bearing in mind Fig. 3 and the results of Guijarro (1986) and Jansà (2014), that the water balance
of the airport cannot, in any way, be extended to the whole island. The water balance shown is
representative of the south of Mallorca. It is also indicative of the situation in the western and
eastern coastal zones and in the center of the island, although the latter zone tends to store a
little more of water in the soil during the winter as consequence of the higher precipitation
(recall Fig. 3). For the northern and northeastern zones of Mallorca the water balance is
expected to be much more similar to that at the Menorca airport, as the rainfall regimes are
quite similar in monthly distributions and amounts. In the mountainous area of Mallorca the
water balance is certainly very different to that at the airport, as the climatological annual
precipitation is almost four times greater. In this zone there are two reservoirs dedicated to the
supply of water to the population, which of course rely on the regular runoff of the fall and
winter. In any case, some drought also exists on the mountains during the summer, since
precipitation in this season is basically absent as in low lands.
In order to validate the previous water balance in terms of precipitation and potential
evapotranspiration, the results of a more sophisticated method have been examined. The data
provided at the web site https://wci.earth2observe.eu/portal/ (which collects data from the
European Earth2Observe project) at the three grid points (resolution 0.25°) closest to the
airports of Mallorca, Menorca and Ibiza have been obtained. Monthly total precipitation values
(PCP) for each grid point have been extracted from 1981 to 2010 and the mean monthly values
have been computed (these monthly data originally come from the analysis performed by Beck
et al. (2017)). Monthly total values of evapotranspiration (EVT; i.e. surface evaporation,
interception and transpiration) and monthly total runoff (R; i.e. surface runoff, sub-surface flow
and deep percolation) provided by the 8 available models have also been obtained from the
Earth2Observe web site. For each model and variable, the mean monthly values with reference
to 1981-2010 have been calculated. Finally, the 8 models-ensemble mean and inter-model
standard deviation of the previous monthly values were obtained. With these values the water
balance was estimated at each of the three mesh points considered. This balance is built as the
precipitation minus the actual evapotranspiration minus the losses (WB = PCP-EVT-R).
These results reveal:

a) For the mesh point near the Mallorca airport (components of the water balance and the
balance itself are displayed in Fig. 7) the precipitation values used by the models are much higher
than those observed at the airport. As an example, the observed mean annual value (1981-2010)
is 411.3 mm, while the same rainfall product used by the models is 597.4 mm. Regarding EVT,
the model ensemble mean values are higher than those obtained at the Mallorca airport for PET
using the Thornthwaite formula, especially in summer. The monthly standard deviations are very
high (that is, large differences among the different models). The PET for the Mallorca airport lies
within the ensemble spread region. Regarding the water balance, and accepting 100 mm as
saturation threshold for the soil, saturation in the Earth2Observe data is obtained during
December, January and February that may be due to the high precipitation values ingested in
the models. Dryness is obtained in July and August and very low water reserve values in June

and September. In the former results a remarkable water deficit is obtained in September (Fig 6), because the temperatures are still high.

b) For the grid point near Menorca airport (not shown), the monthly values of precipitation used by the models are much more similar to those observed at the airport (registered annual average of 548.6 mm versus 601.2 mm in the models). The EVT shows a behavior similar to that at the grid point near the airport of Mallorca: values are greater than those of PET obtained from the Menorca airport data using the Thornthwaite expression, and there is a large spread among the 8 models. The calculated PET values are also well encompassed by the ensemble dispersion band. Regarding the water balance, saturation of the soil is obtained in January and February and a value close to saturation in December. Dryness is also obtained in July and August. These results are very similar to those obtained directly in our study for the Menorca airport.

c)      For the grid point close to the Ibiza airport (not shown), the average monthly precipitation values used by the models are also significantly higher than those registered at the Ibiza airport (observed annual mean of 411.1 mm versus 497.2 mm in the models). Again the average EVT values of the ensemble are larger than PET values given by the Thornthwaite's expression at the Ibiza airport. The inter-model spread is very high. Regarding the water balance, saturation of the soil is not reached in any month; in contrast dryness is present during May, June, July and August. These results are in agreement with the results obtained directly with the airport data.

In conclusion, it seems that the simple method used in the paper is sufficient to obtain a clear representation of the drought object of the study.

## 3.   Hydrologic Year 2015-16.

As already mentioned, the hydrological year 2015-16 was characterized by a negative anomaly with respect to the reference period (Fig. 5). Other hydrological years exhibit greater negative anomalies, but it was the widespread deficit of precipitation during 2015-16 what characterizes the hazardous effects of this drought event. Figure 8 presents the hydrological balance for Mallorca, Menorca and Ibiza airports corresponding to that hydrological year. For these water balances, daily potential evapotranspiration has been calculated using the Hargreaves method (Hargreaves and Samani, 1985). The monthly values have been obtained from the daily values. The distribution of rainfall shows significant accumulations in September, due to the convective rains that affected the islands (176.4 mm in Mallorca, 181.1 in Menorca and 139.6 in Ibiza). It is also evident the quite low rainfall recorded during the rest of the hydrologic year, particularly during the rest of the autumn and the whole winter. At Mallorca airport the precipitation during November 2015 to January 2016 was 25.6 mm, which represents the lowest value among the 43 considered hydrologic years (Fig. 9). Similarly, the total precipitation recorded during December 2015 was 0.2 mm, the lowest of this month for the whole period 1973-2016. At Menorca airport the accumulated precipitation from November 2015 to January 2016 was 45 mm, also the lowest quantity recorded in a hydrologic year. The precipitation for December 2015 was 2.1 mm, again the minimum record for this month during the period 1973-2016. In Ibiza the situation was similarly extreme, since 35.2 mm was the precipitation recorded for

November-January, the lowest for the 43 analyzed hydrologic years, and only 0.7 mm were registered in December 2015 (only surpassed by the 0.2 mm recorded in 1974).

For the Mallorca airport (Fig. 8) it is observed that already during the month of October, the water that was stored in the soil as consequence of the heavy precipitation events of September, was already consumed; during the rest of the year there is deficit. The lack of precipitation during the winter months implies a very dry soil when the sunny days and rise of temperatures establish in spring.

Something similar happens in Ibiza, where the water deficit starts a bit later than in Mallorca as a consequence of the rainy early autumn (September and October) but where the abnormal lack of winter rains is also quite remarkable. In Menorca the situation is to some extent similar: the deficit begins in March, although the winter precipitation was also very scarce.

The Thornthwaite approach applied for obtaining PET monthly climatic values uses directly the monthly mean temperatures provided by AEMET for the period 1981-2010. The aim is to build a reference water balance for a comparison with the particular water balance of the hydrological year 2015-16. For this hydrological year, the PET monthly values have been calculated from the daily values obtained by the Hargreaves method. Some comparison between the two methods for this year is necessary to fully justify the reference to the climatic water balance. A comparison between both methods was made. Specifically, monthly PET values using Thornthwaite method were calculated for the hydrological years 2014-15 and 2015-16 at the E1 site in Mallorca (see Fig. 1). Analogous monthly values were obtained from the daily PET values given by the Hargreaves formula. The E1 station is located in the most arid region of the island. The two time series show a correlation coefficient of 0.9 (see Fig. 10). For the warmer/colder months the Thornthwaite method reveals larger/lower monthly PET values than the other approach. In any case, given the high value of the correlation coefficient, the obtained reference or climatic water balance can be effectively compared with the one calculated for the 2015-16 hydrological year.

Comparing the water balances of 2015-16 (Figure 8) with the climatic water balances (Figure 6) at the three airports, notable differences during the fall and winter are found. In the climatic balance the beginning of autumn shows a water deficit that is rapidly reversed during the rest of autumn and winter. Winter rains develop the reserves for the ground, since the summer is extremely dry. Only at the Menorca airport this storage exceeds the 100 mm threshold and therefore surface runoff and infiltration are produced. The lack of rainfall in the Balearic Islands, especially during the extreme winter of 2015-16, gives an idea, when analyzed in terms of the water balance, of the hydrological stress to which the local vegetation was subjected. This deficit of precipitation during the winter in the Mediterranean area has been related to some more general droughts observed in Europe (Vautard et al. 2007).

It is interesting to display some other areas of Mallorca that were affected by a still more intense drought, again in terms of their water balances. Figure 11 shows the water balance for 2015-16 obtained from the data at three automatic meteorological stations located in the south, central and northern parts of Mallorca (see Fig. 1). It can be observed that at the southernmost station (E1) the precipitation throughout the year was lower than the potential evapotranspiration, motivating that the water deficit was accumulating during the whole hydrological year. The intense rains that affected the airport location in September did not occur in this area. The lack of precipitation in winter is remarkable. The accumulated drought that reached the always dry summer was very severe and had dramatic consequences on the vegetation types possessing

shallow roots. But also on some trees, especially almond trees, whose fruit maturation had to
develop under absolutely unfavorable conditions.
The precipitation regime in the north of Mallorca (E3) was very similar to that of the south
region. Rainfall was also lower than potential evapotranspiration during all the months of the
hydrological year. In the center of the island (E2) the situation was not very different, although
during the month of September the precipitation was enough to surpass the potential
evapotranspiration. The rainfall and the evaporation regimes resemble those at the airport. The
convective rains of early fall also reached the center of the island, but the profound lack of
rainfall in winter was a constant that is repeated at all locations, supposing that evaporation
rates permanently exceed precipitation, a feature clearly divergent from what is climatologically
expected.
The hydrological year 2015-16 was characterized by very intense rainfall events in September
followed by a persistent lack of rainy situations for the rest of the period. This begs the question
of the role of runoff, especially when the season starts with heavy precipitation, on soil dried
out by the summer; in these conditions much less water will infiltrate and thus recharge soil
moisture. There is an added problem for an accurate computation of the water balance when
measurements on the runoff are not available, as it is the case of our study. There are very few
measurements of runoff in streams of the Balearics, all of them belonging only to special
campaigns and always before 2014. Note that in the Balearic Islands there are not permanent
rivers. In addition, no information about the episode can be obtained from the Earth2Observe
web page, since model data extends only till 2012.
However, the runoff coefficient for a nearby stream basin to the Mallorca airport (few
kilometers away) was estimated by García et al. (2017), based on observed stream flows for the
1977-2009 period. The estimated runoff coefficient was as low as 0.03. This result ensures that
the conversion of precipitation into surface runoff is quite low for this nearby basin.
Furthermore, no substantial changes are found in the spatial distribution of the physiography
and hydrology of the stream basin where the meteorological station is located. We can safely
assume that almost all precipitation is infiltrated and that the P-E balance is quite realistic when
assessing the climatic water balance for the 1981-2010 period. Even for the heavy precipitation
event at the end of the 2015 warm season, most of precipitation would have infiltrated owing
to the high infiltration capacity of the soil and its low water content.

**4.- Circulation patterns.**

During the winter of 2015-16 the North Atlantic was especially active cyclonically speaking.
Many deep depressions developed above $45^0$-$50^0$ of latitude and affected Europe. Particularly
deep was the impact on Ireland and England, especially in December, where very intense rains
(up to 200% of the climatic value referred to 1981-2010 for that winter (McCarthy et al., 2016))
resulted in floods. The substantial westerly flow also advected warm air along that latitude belt
and the mean winter climatic temperature values (period 1981-2010) were largely exceeded in
Ireland and England, up to two degrees in the south of England. In December this warm anomaly
in the south of England reached 5 degrees (McCarthy et al., 2016). This situation was caused by
a strong zonal circulation of the jet stream over the North Atlantic; the jet basically pointed
directly to Ireland from the coasts of America during that winter (Burt and Kendon 2016). For
latitudes below 50$^0$, the westerly flow was also maintained during that winter. Figure 12 shows
the average geopotential structure at 500 hPa for Europe and the Mediterranean for November
2015 to January 2016. High geopotential values over the Iberian Peninsula and the western
Mediterranean that extend towards Central Europe are evident. For these months the NAO
index was 3.56 for November, 4.22 for December and 1.16 for January
(https://crudata.uea.ac.uk/~timo/datapages/naoi.htm), thus reflecting a strong westerly
circulation.
The above meteorological situation is unfavorable for any significant occurrence of rainfall in
the western Mediterranean and particularly in the Balearic Islands. The most favorable rainfall
conditions in the islands are linked with the evolution of cyclonic disturbances at mid-upper
tropospheric levels which give rise to secondary depressions at surface over the Mediterranean
and easterly moist flows impinging over the Balearic Islands (Romero et al. 1999). Atlantic
disturbances crossing central Europe, even involving active fronts, generally produce little
precipitation along the Spanish Mediterranean coast and in the Balearic Islands, in any case just
affecting the northern half of the islands. Figure 13 shows that during the months of November
2015 to January 2016, when the precipitation in the Balearics was practically null, there was a
strong positive anomaly of geopotential at 500 hPa over the western Mediterranean, a
circulation pattern entirely inhibiting the generation of any type of precipitation system.
It was previously reported that during September 2015 intense precipitation happened on all
three islands. Figure 10 shows that the atmospheric circulation during this month was
characterized by the presence of lows at 500 hPa, indicated by the nucleus of negative anomaly
affecting Western Europe and the Western Mediterranean. This pattern is dynamically favorable
for the generation of heavy rainfall situations slightly downstream, over the Spanish
Mediterranean coast and the Balearic Islands (Romero et al., 1999).
The average conditions displayed in Figure 13 show the radical change of the circulation that
occurred between September and November 2015. The pattern of September would
correspond, at low levels, with the persistence of meridional flows over the north Atlantic and
low NAO values (-1.65 for September and -1.13 In October), the opposite pattern found during
the period from November 2015 to January 2016. The occurrence of rainfall in the Balearic
Islands could be better correlated with high values of the Scandinavian Index (September 1.09,
October 0.62, November -1.4, December 0.08, January -0.68, normalized to the period 1981-
2010; http://www.cpc.ncep.noaa.gov/data /teledoc/scand.shtml).
As a contrasting situation, the hydrological year 2008-09 can be considered a wet case (see
Figure 5). During the months of November to January, 214 mm at the Mallorca Airport, 303 mm
at the Menorca Airport and 187 mm at the Ibiza Airport were recorded. Figure 11 shows the
geopotential anomaly at 500 hPa from November 2008 to January 2009. A notable negative
anomaly centered over the western Mediterranean can be observed, resulting in a completely
opposite pattern to that of 2015-16 (Figure 14). The values of the NAO index for these months
were negative or low (November -1.30, December -0.58, January 0.6).

**5. Conclusions.**
The characteristics of the recent drought that occurred in the Balearic Islands during the 2015-
16 hydrological year (September to August) have been presented. The analysis was carried out
in terms of the particular hydrologic balance for this year using data from six meteorological
stations to determine the potential evapotranspiration and to estimate the actual evaporation.
These water balances have been compared against those corresponding to the long-term
climatic conditions for the reference period 1981-2010. Comparison of the climatic water
balance calculated with the empirical expressions against the balance deduced from 8 models
used by the European Earth2Observe project show some differences. Most of these differences
have to be attributed to the greater values of precipitation ingested in the models and the high
variability of the simulated evaporation and runoff. However, the calculated values of PET lie
within the spread interval of the models.

The analyzed hydrologic year reveals a profound precipitation deficit during the winter, such
that the potential evapotranspiration surpassed the precipitation practically the whole year,
except in September when at some stations the precipitation exceeded the evaporation. The
recorded precipitation from November 2015 to January 2016 was the lowest for this period at
the three airports of the Balearic Islands for the 43 considered hydrologic years. The
precipitation of December was also unappreciable in all three islands. Accordingly, the soil could
not store any water to face the spring, when insolation hours and temperatures increased. This
resulted in the lack of any water reserves during 2015-16, an aspect totally anomalous compared
with an average winter, for which certain levels of moisture can be maintained in the soil until
June in Mallorca and Ibiza, and until July in Menorca.

We verified that the meteorological situation during the anomalous 2015-16 winter was
dominated by a very marked westerly flow over the North Atlantic, with high values of the NAO
index. This situation caused intense precipitations and anomalously warm temperatures in
Ireland and England. On the contrary, precipitations at lower latitudes, and particularly in the
western Mediterranean, were very scarce.

The identification of anomalous circulation patterns in seasonal or climate prediction models
can be a mechanism for anticipating drought situations and stimulate planning and mitigation
measures in a region like the Mediterranean, where water demand is high, especially at the
time of the year when precipitation is scarce. It is also a promising line of research for purposes
of agricultural planning and conservation of the current vegetation.

**Acknowledgments**
Temperature and precipitation data were recorded and provided by the Spanish Meteorological
Agency (AEMET). The weather analyses correspond to the NCEP/NOAA reanalysis database
(https://www.esrl.noaa.gov/psd/cgi-bin/data/composites/printpage.pl). Figure 3 comes from
the PREGRIDBAL project (http://pregridbal-v1.uib.es/). References to media correspond to
Diario de Mallorca. The authors acknowledge the reviewers for their constructive comments
that contributed to improve the original version of the paper. This research was sponsored by
CGL2014-52199-R (EXTREMO) project, which is partially supported with FEDER funds, an action
funded by the Spanish Ministerio de Economía y Competitividad.

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

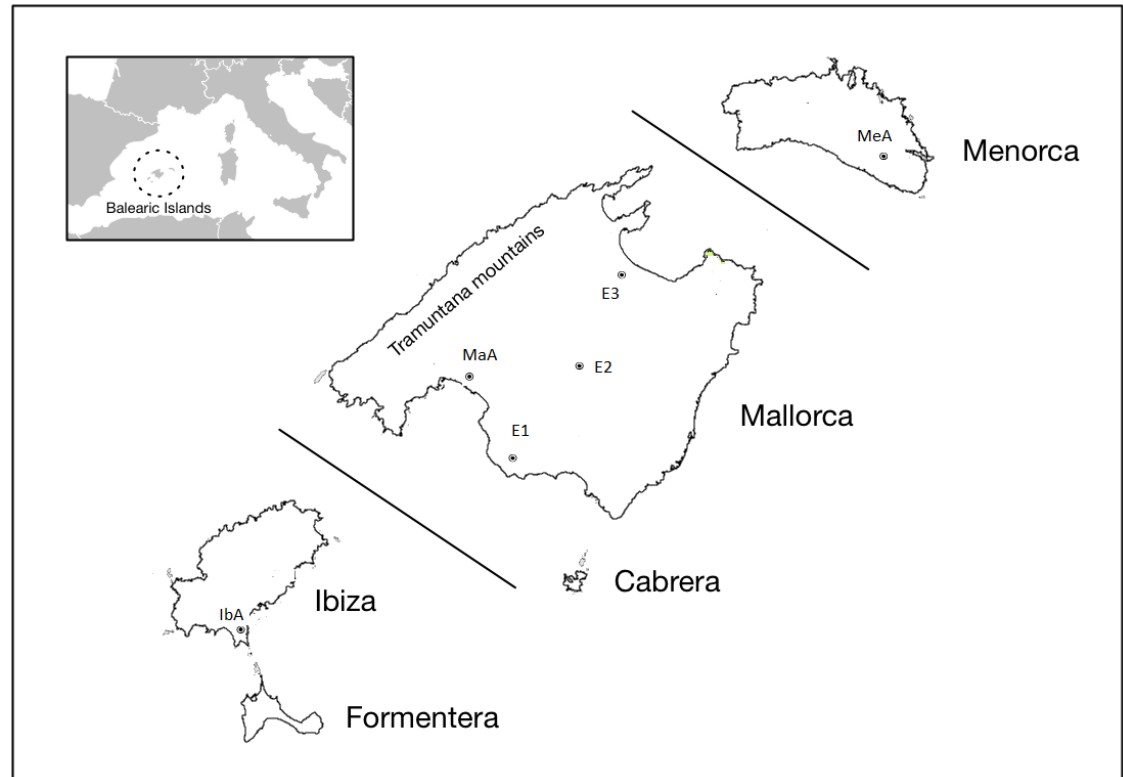



Figure 1.- The Balearic Islands. MaA, Mallorca airport; MeA, Menorca airport; IbA, Ibiza airport.
Locations of the other climatological stations analyzed in the text are also indicated.
















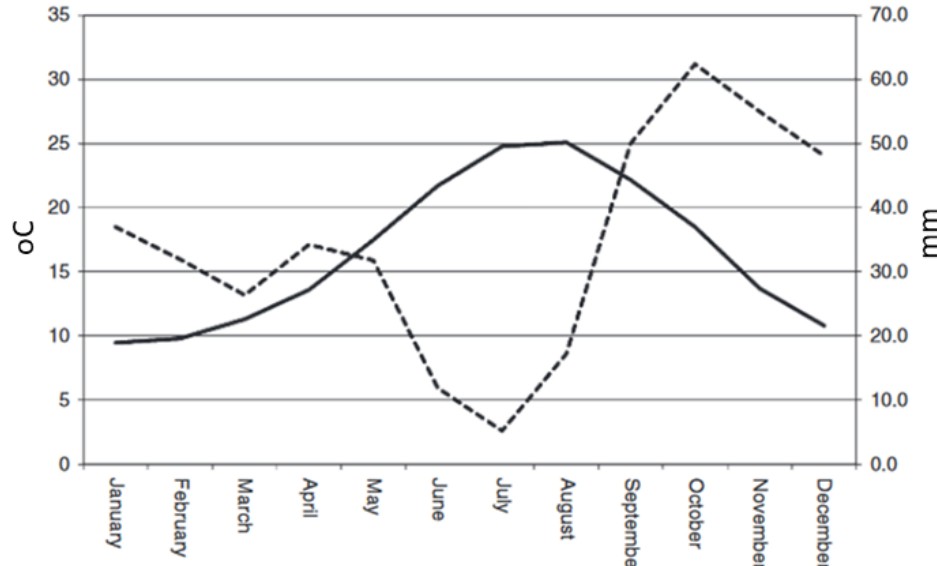


Figure 2. Ombrothermic diagram (Gaussen 1955) for Mallorca airport (1981-2010) (after Jansà
et al., 2016). Continuous line: mean temperature. Dashed line: mean precipitation.








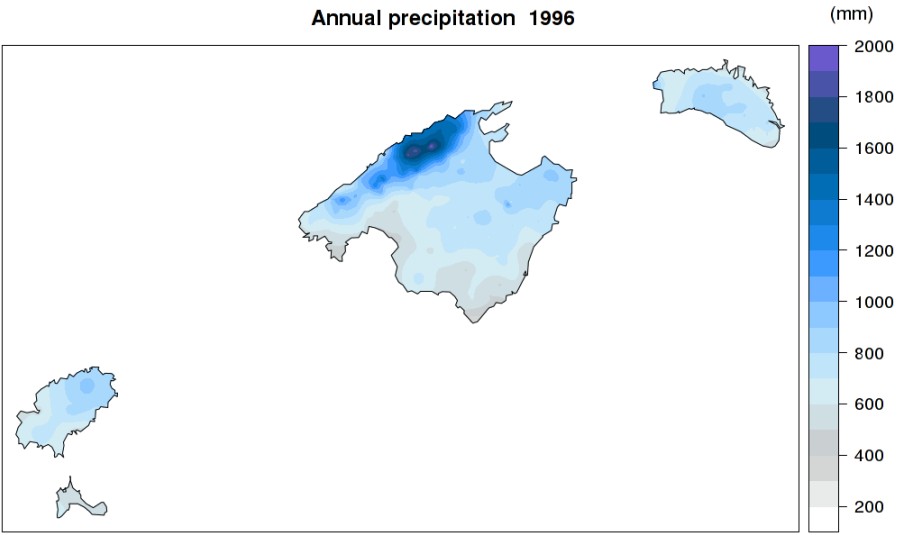


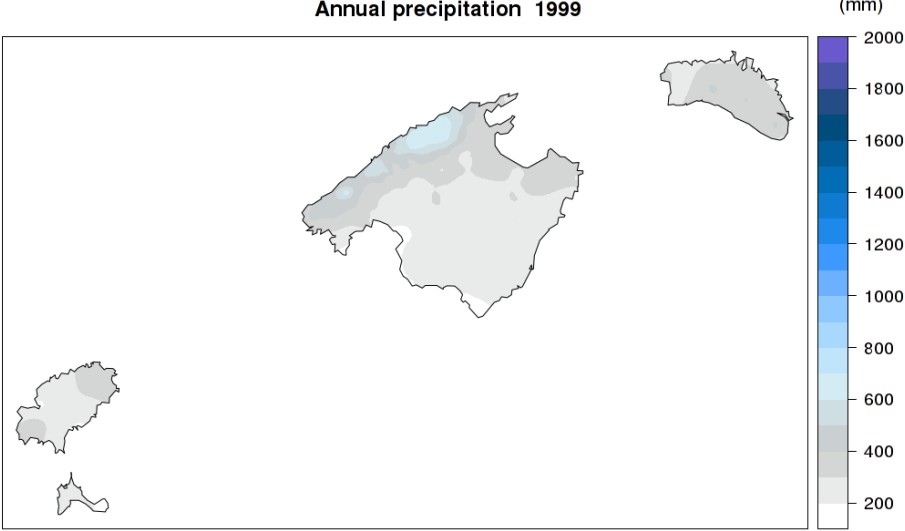


Figure 3.- Spatial distribution of accumulated precipitation for 1996 (wet year) and 1999 (dry year). The same scale is used. (from http://pregridbal-v1.uib.es/).







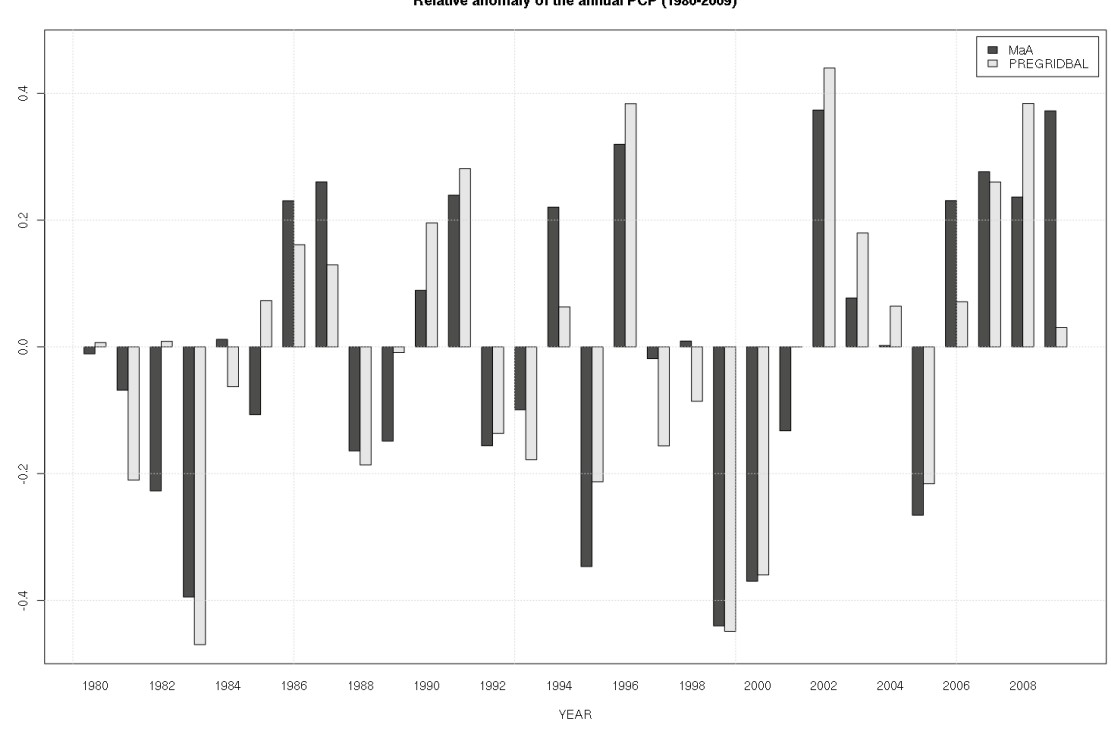



Figure 4.- Time series of the relative annual precipitation anomalies at the Mallorca airport and
for Mallorca as a whole derived from the PREGRIDBAL project.

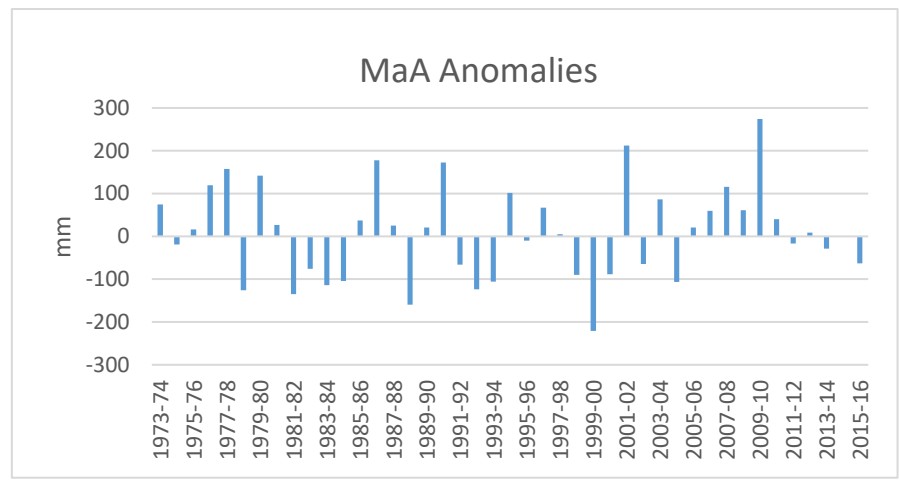

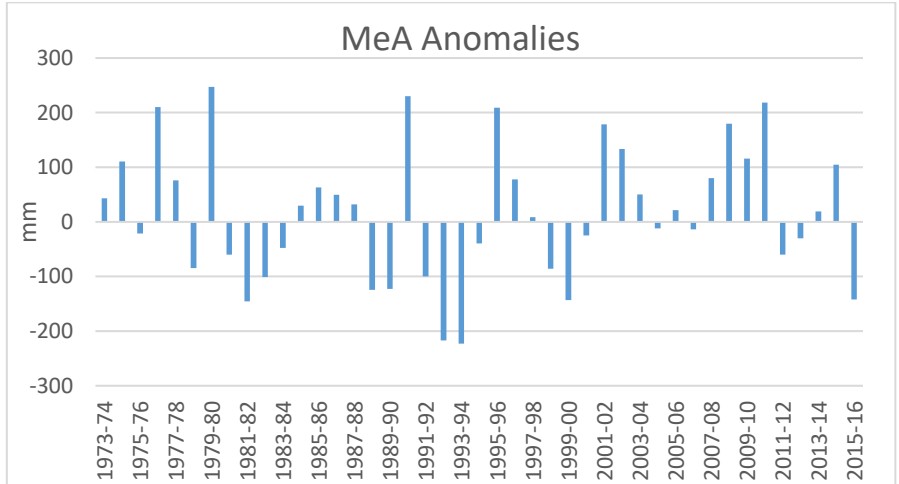

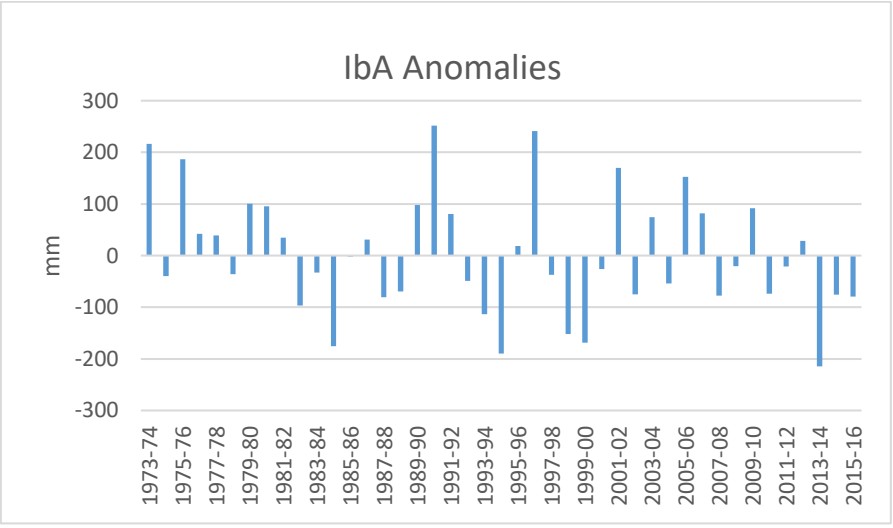





Figure 5.- Anomalies of the precipitation for the hydrological year at the airports of Mallorca
Menorca and Ibiza with respect to the respective averages calculated for the reference period
1980-81 to 2009-10.



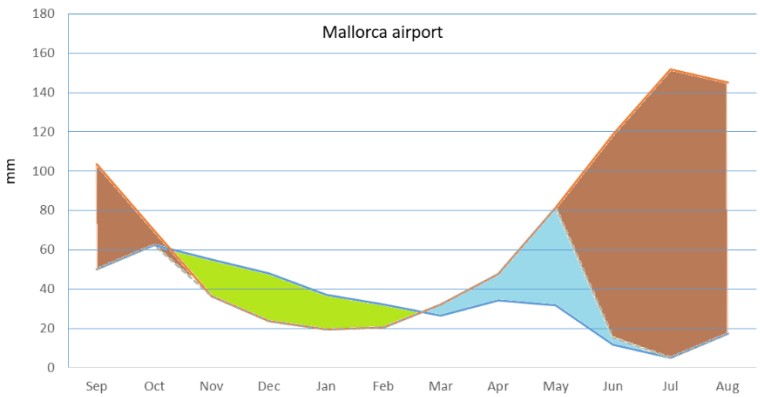


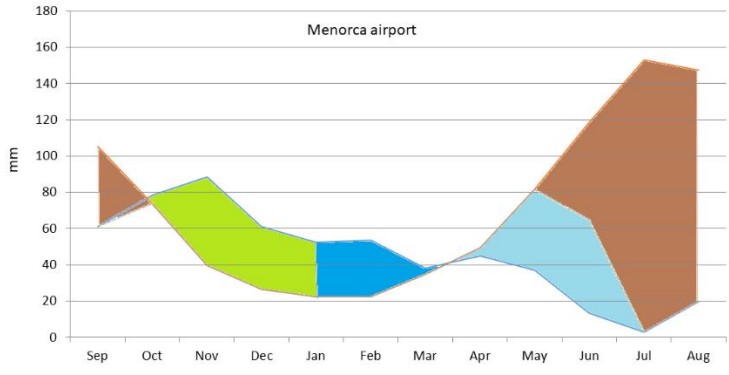


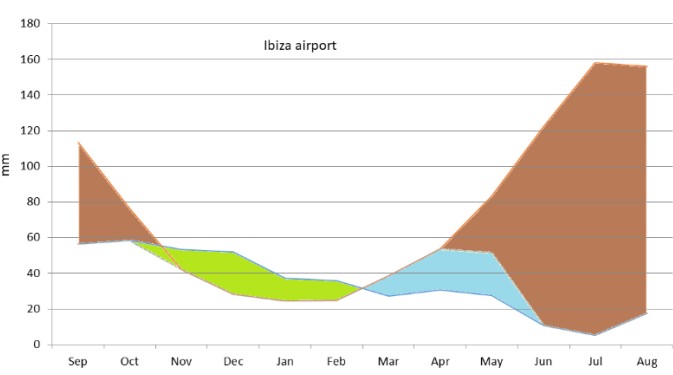


Figure 6.- Climatic water balance (1981-2010) at the airports of Mallorca, Menorca and Ibiza (MaA, MeA and IbA in Fig. 1). Lines indicate: blue, precipitation (mm); brown, potential evapotranspiration (mm); dashed green, evaporation (mm). Colored areas indicate: green color, accumulation of water in the soil; cyan, evaporation of water stored in the soil; blue, runoff; brown, water deficit in the soil.




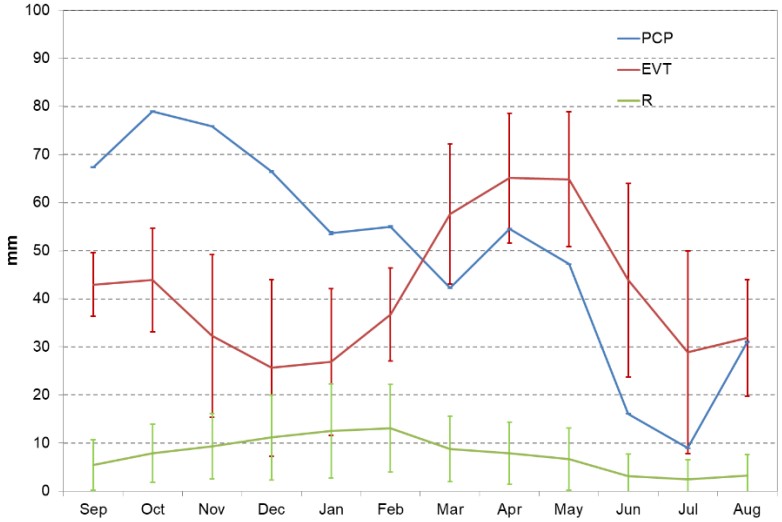



Figure 7.- Top: Components of the climatic water balance (1981-2010) at the grid point nearest
to the Mallorca airport, deduced using data from the European Earth2Observe project. Vertical
bars represent standard deviation among the 8 models. Bottom: Water balance at the same grid
point (see text for details).


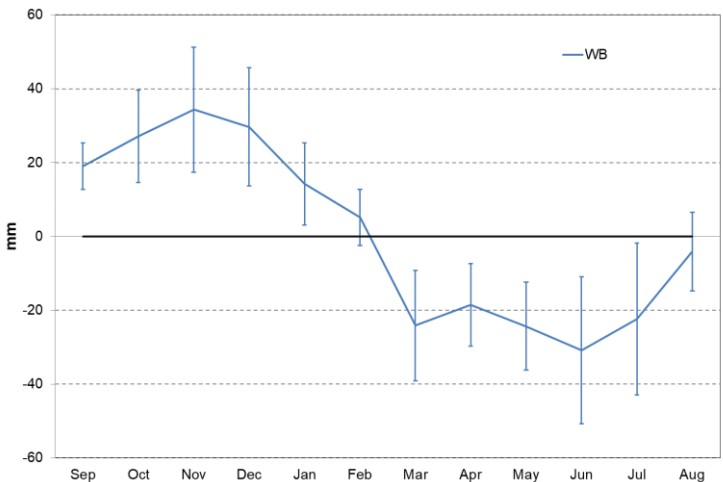

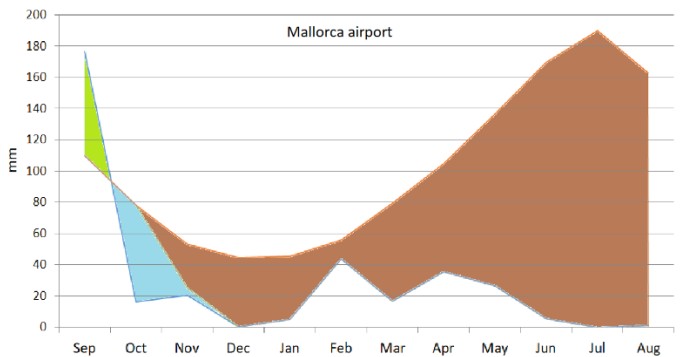


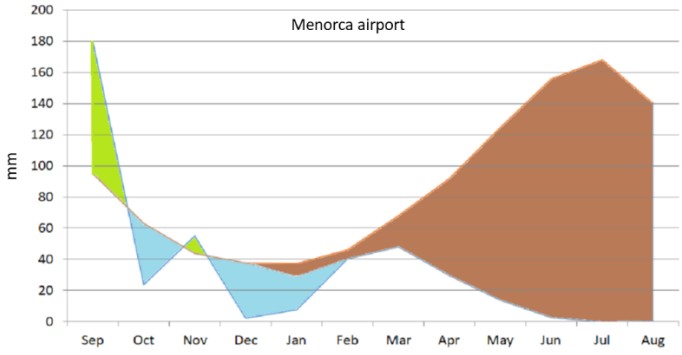


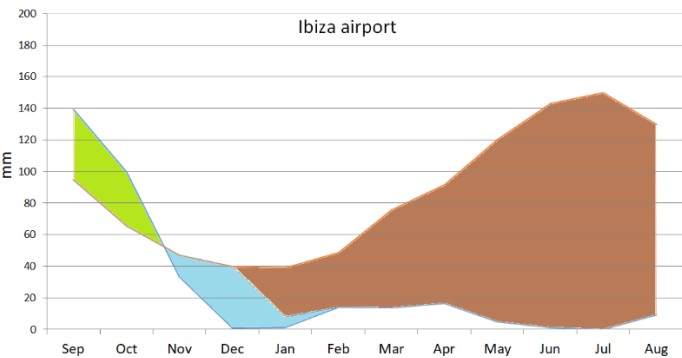


634          Figure 8.- As in Figure 6 but for the hydrologic year 2015-16.












Figure 9.- Accumulated precipitation from November to January at the Mallorca airport.
















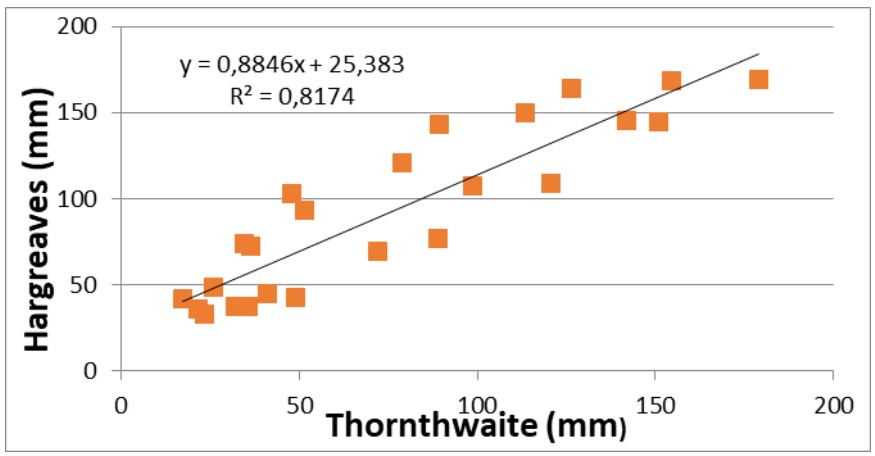



Figure 10.- Comparison of the monthly PET obtained by the Thornthwaite and Hargreaves
methods at E1 station (see Fig. 1) for the hydrological years 2014-15 and 2015-16.















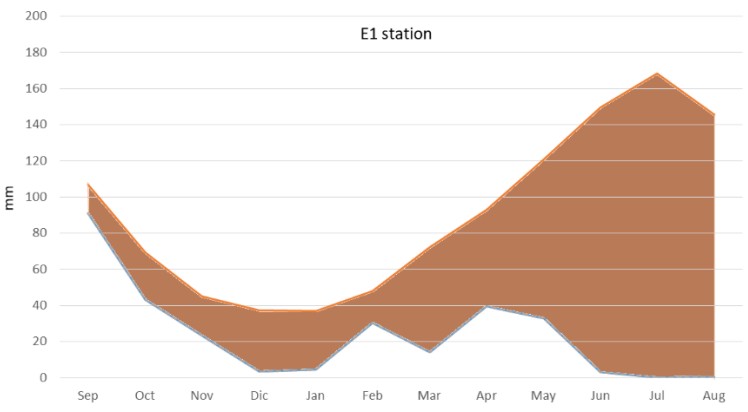


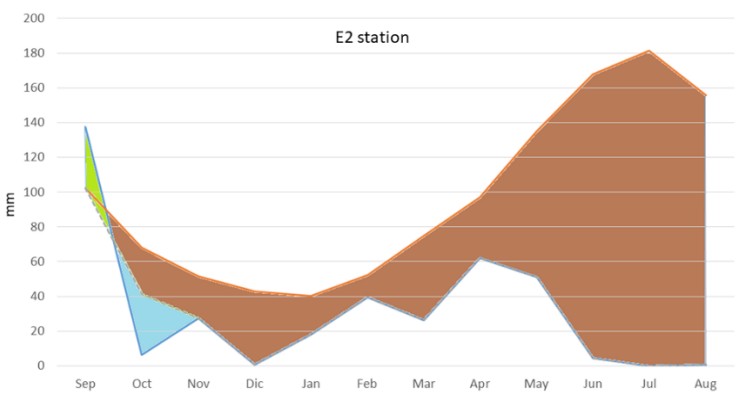


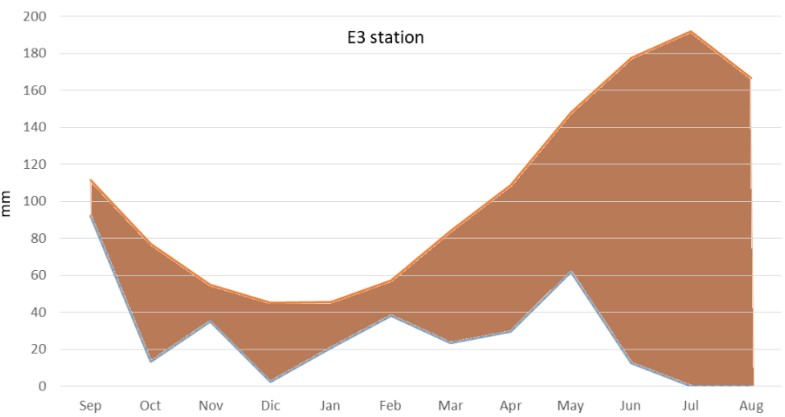



Figure 11.- As in Figure 8 but for the three additional locations in Mallorca (E1, E2 and E3 in
Figure 1, respectively).

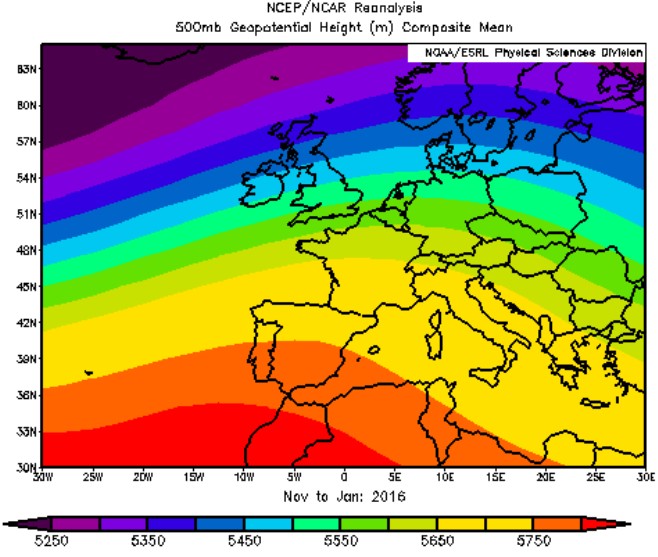



Figure 12.- Mean geopotential height at 500 hPa for November 2015 - January 2016 (source
NCEP/NOAA reanalysis)

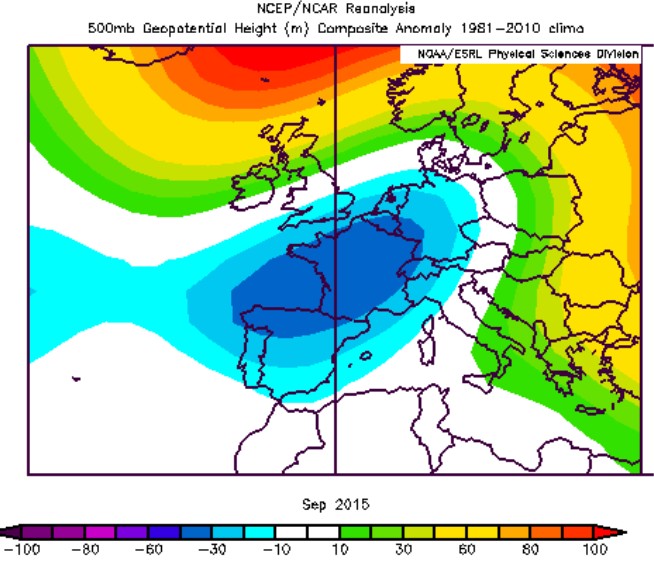



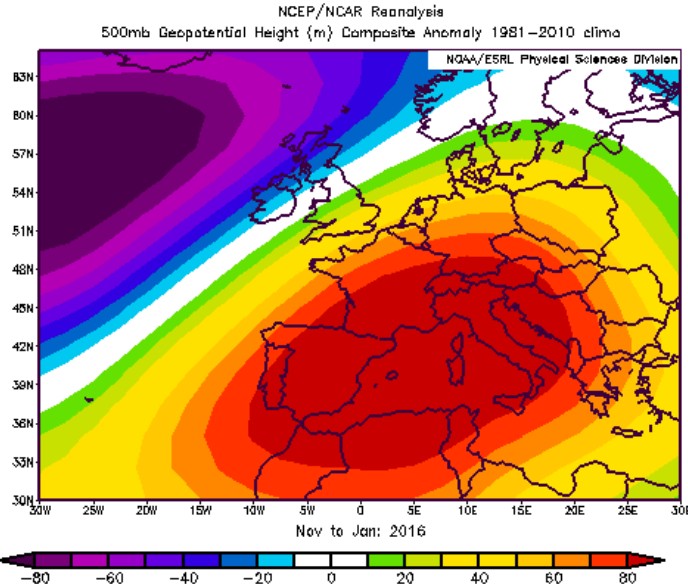



Figure 13.- Geopotential height anomalies at 500 hPa for September 2015 and for November
2015 - January 2016, referring to the reference period 1981-2010. (source NCEP / NOAA
reanalysis)



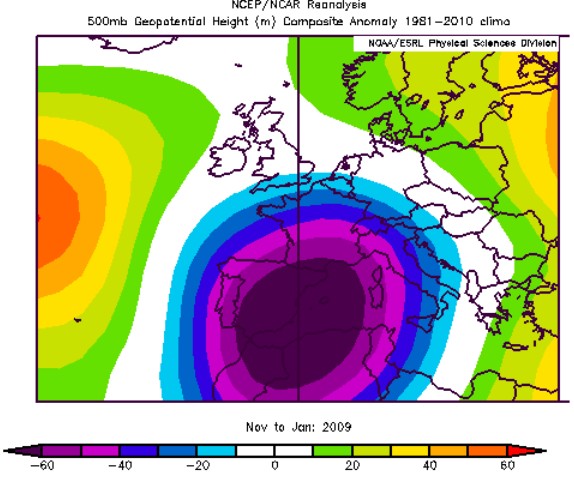



Figure 14.- Geopotential height anomalies at 500 hPa for November 2008 - January 2009 with
respect to the reference period 1981-2010. (source NCEP/NOAA reanalysis).

