# Peer review of "On the drought in the Balearic Islands during the hydrological year 2015-2016"

_Natural Hazards and Earth System Sciences, 2017_

## Referee Comment (RC1) · Anonymous Referee #1 · 23 Jul 2017

This paper on the extreme droughts of 2015-16, or more generally on the water resources, of the Balearic islands covers a rather important topic. These islands, like the rest of the world, faces a changing climate but they are at the same time in already quite arid conditions and furthermore they draw their resources from a very water demanding industry, namely tourism. Thus I would like to recommend NESS to consider for publication as it falls well within the scope of the journal. But in my opinion some issues need to be resolved in this manuscript before publication. I would like to encourage the authors to take advantage of this review to put their study on a stronger footage.

Major comments :

a) The entire study hinges on the meteorological stations of the airports of the three

island. Which I guess is fine as these are high quality data sets and provide multiple meteorological variables. Still, as in the semi-arid climate rainfall is sparse (by definition !) and very variable in space, it is not obvious that a dry year at the airport is also dry on the entire island. Thus I think that the study would benefit from an analysis of the spatial representativness of the inter-annual variability of the rainfall captured at the airports. Using all the rain gauges of the islands (or the gridded data displayed in figure 4) one could estimate the error done on the characterisation of the inter-annual variability with only the airport stations.

b) The usage of empirical formulas for estimating potential evapo-transpiration (PET) is understandable in view of the data situation. But the authors should point out that these empirical formulations have their limits, especially in a changing climate. There is wealth of literature on these aspects. Furthermore it is not very stringent to use in section 3 for the validation of the methodology the Thornthwaite formulation, and then later for the hydrological years 2015-16 Hargreaves. There should at least be some comparison of these methods and proof that the choice of empirical function does not affect the conclusion of the study.

c) The analysis of the drought in terms of a simple P-E analysis is interesting but is based on some crude assumptions. Thus it is important to bring some arguments as to why these are justified in the present case. The community has now some powerful tools to analyse and predict the continental water cycle and it is a petty that there is no attempt to show what is captured and what is lost in the proposed approach. I would recommend the authors to look at the excellent continental water balance re-analysis which has been produced by the European Earth2Observe project : https://wci.earth2observe.eu/portal/. As this work is performed with multiple models and at 1/4° degree resolution, there is some interesting information to be drawn for this study of the Balearic islands. d) The hydrological year 2015-16 is characterized by very intense rainfall events in September and then a continuous deficit for the rest of the rainy period. This begs the question of the role of runoff. Especially when the

season starts with heavy precipitation, on soil dried out by the summer, much less water will infiltrate and thus recharge soil moisture. An analysis of the Earth2Observe multi-model ensemble or river discharge data could offer some more insight into the other terms of the water balance equation. It would shows how far the simple P-E balance can be trusted. Furthermore the authors use PET and E, so there is a further assumption to be tested here.

Minor Comments :

1. I guess it is OK to reference general media in a scientific publication in order to highlight the societal relevance of the topic studied. But this should not exempt the authors from putting proper references. So please cite the newspaper which served as a source of the information.

2. 3rd paragraph of Introduction : "somehow characterizes the type of natural vegetation". This seems very vague especially when you consider that there are vegetation classification which could be used to refine such a statement.

3. Is it necessary to repeat the inter-annual analysis for the civil and hydrological years (figures 3 and 5) ? As pointed out above the spatial representativeness seems more critical to me.

4. It is unclear for which year figure 6 is ! This is neither explained in the text nor the caption of the figure.

5. Many figures do not have units on the axis or title and we have to retrieve that information from he caption. In general the figures are of poor quality.

6. Description of figure 8 in the text and its caption do not correspond. Are three stations used or only the airport data ?

7. End of conclusion : The natural vegetation in the Balearic islands is not wild I think !!

---

## Referee Comment (RC2) · J.A. Guijarro (Referee) · 8 Aug 2017

General comments:

I agree with referee #1 on the importance of the subject and the convenience of the publication of this work.

It is also clear the need to use empirical formulas to calculate evapo-transpiration rates, due to the lack of direct measurements. In this sense, the Hargreaves method is much used when the variables observed at a site are limited. However, the airports are observatories with a more complete range of observations, and therefore more complex alternatives, such as the Penman-Monteith method, could have been used. Unfortunately, radiation is measured only at the Palma airport and, while a comparison on ET

values computed with Penman-Monteith and Hargreaves formulas would assess the validity of the latter within the studied climatic area, that effort could give birth to a new article by itself.

More debatable is, as referee #1 also points out, the use of both Thornthwaite monthly hydrological balances and daily balances with the Hargreaves estimates. Thornthwaite approach has been extensively used in the past when dealing with monthly series, and can still be used for comparison purposes, especially on monthly climatic averages. However, it has been reported that Thornthwaite method under-estimates ETP in arid climates, but if more realistic empirical formulas are used (Guijarro, 1986, cited by the authors), monthly hydrological balances following Thornthwaite result in soils being completely dry all year round in extensive zones of the Balearic islands, which is also unrealistic. Therefore, climatic balances should be derived by applying daily balances to the reference period and then computing the monthly values, or at least the Thornthwaite method should also be applied to the 2015-16 year to assess the impact of using these different approaches.

Specific comments:

115: "These are the longest homogeneous climatic series without gaps..." Has their homogeneity been assessed? I would remove ' homogeneous' otherwise.

With respect to comments on lines 132-136 and 155-156 a reference could be added to the sentence in line 156: "do not respond to the same circulation patterns, as previously reported by Guijarro (2002 and 2003)" GUIJARRO JA (2002): Tendencias de la precipitación en el litoral mediterráneo español. In Guijarro et al. (Eds.), El agua y el clima, Asociación Española de Climatología, A-3:237-246, ISBN 84-7632-757-9. GUIJARRO JA (2003): El flujo geostrófico superficial en el Mediterráneo Balear durante el periodo 1948-2002. Rev. climatol., 3:45-59.

The statement in lines 137-138 is debatable. Different regimes can be seen in different parts of Mallorca (Sumner et al., 1995). SUMNER G., GUIJARRO J.A., RAMIS

C. (1995): The impact of surface circulation on significant daily rainfall patterns over Mallorca. International Journal of Climatology, 15:673-696.

Technical corrections:

131: 'end' → 'and'

254: "to which the local vegetation was subjected to." Too many 'to's? Remove the last one?

---

## Author Comment (AC1) · 26 Sep 2017

**Reply to referee 1**

*This paper on the extreme droughts of 2015-16, or more generally on the water resources, of the Balearic islands covers a rather important topic. These islands, like the rest of the world, faces a changing climate but they are at the same time in already quite arid conditions and furthermore they draw their resources from a very water demanding industry, namely tourism. Thus I would like to recommend NESS to consider for publication as it falls well within the scope of the journal. But in my opinion some issues need to be resolved in this manuscript before publication. I would like to encourage the authors to take advantage of this review to put their study on a stronger footage.*

Thank you very much for the detailed review of the manuscript. Your major and minor comments will be very useful to produce an improved version of the paper.

*Major comments :*

*a) The entire study hinges on the meteorological stations of the airports of the three island. Which I guess is fine as these are high quality data sets and provide multiple meteorological variables. Still, as in the semi-arid climate rainfall is sparse (by definition !) and very variable in space, it is not obvious that a dry year at the airport is also dry on the entire island. Thus I think that the study would benefit from an analysis of the spatial representativeness of the inter-annual variability of the rainfall captured at the airports. Using all the rain gauges of the islands (or the gridded data displayed in figure 4) one could estimate the error done on the characterisation of the inter-annual variability with only the airport stations.*

As described in the article, the spatial variability of the mean annual rainfall in Mallorca is very high. It is thus not so clear that the time series at the Mallorca airport captures the interannual variability of precipitation across the island.

Following the idea of the reviewer, we have conducted an analysis of the spatial representativeness of the interannual variability captured by the airport of Mallorca using two methodologies. First, the time series of the relative annual anomalies (anomaly divided by the corresponding annual average) have been calculated for five meteorological stations located in Mallorca, and the resulting mean time series (of the five individual series) has been determined. The five stations are representative of different pluviometric regimes of the island: mountainous area, north, center, east and south. For this analysis, the period 1981-2010 has been considered. The time series of annual relative anomalies at Mallorca airport has been compared against the above mean time series. Both time series have a correlation coefficient as high as 0.9.

The second method is analogous to the previous one but uses the precipitation analyses across the island of Mallorca that were derived in the PREGRIDBAL project (López et al 2017). These analyses have a resolution of 100 m and use all available observed data for each product requested. Annual precipitation grid data has been considered for each of the years 1980-2009, together with the grid analysis of mean precipitation corresponding to these 30 years. For each mesh point and for each year the relative annual anomalies have been determined and a time series expressing the spatial average of annual anomalies has been calculated. Finally, this time series has been compared against the relative anomalies at Mallorca airport, yielding in this case

a correlation coefficient of 0.86. Thus, it seems well justified the assumption that the spatial-temporal variability in the island of Mallorca is captured by the series of precipitations at the airport of Mallorca.

Due to their relatively small size and moderate orography, the spatial variability of the annual mean precipitation in Menorca and Ibiza are much lower than in Mallorca, therefore it seems clear that the corresponding time series at the airports are even more representative of the corresponding interannual variability of these islands than in Mallorca

This discussion will be included in the new version of the paper. The following figure shows the result of the second method.

[Figure]

*b) The usage of empirical formulas for estimating potential evapo-transpiration (PET) is understandable in view of the data situation. But the authors should point out that these empirical formulations have their limits, especially in a changing climate. There is wealth of literature on these aspects. Furthermore it is not very stringent to use in section 3 for the validation of the methodology the Thornthwaite formulation, and then later for the hydrological years 2015-16 Hargreaves. There should at least be some comparison of these methods and proof that the choice of empirical function does not affect the conclusion of the study.*

We will introduce a short discussion in the revised version of the manuscript on the limitations of the empirical formulation of the PET.

The Thornthwaite approach has been used to obtain PET monthly climatic values by using directly the monthly mean temperatures provided by AEMET for the period 1981-2010. The objective is to have a reference water balance to perform a comparison with the particular water balance for the hydrological year 2015-16. For this hydrological year, the PET monthly values

have been calculated from the daily values given by the Hargreaves method, which provides better results owing to the use of daily data. We agree that some comparison between the two methods for this year is necessary if some reference to the climatic water balance is made. Although not included in the paper, a comparison of both methods was made during the development of the study. In fact, the monthly PET values using Thornthwaite method were calculated for the hydrological years 2014-15 and 2015-16 for the E1 in Mallorca (see Fig 1 in the original manuscript). The same monthly values were obtained from the daily PET values using the Hargreaves formula. The E1 station is located in the most arid region of the island. The two time series exhibit a high correlation coefficient, 0.9. For the warmer months the Thornthwaite approach provides larger PET values and the opposite for the colder months (see figure).

This discussion will be included in the paper.

[Figure]

*c) The analysis of the drought in terms of a simple P-E analysis is interesting but is based on some crude assumptions. Thus it is important to bring some arguments as to why these are justified in the present case. The community has now some powerful tools to analyse and predict the continental water cycle and it is a petty that there is no attempt to show what is captured and what is lost in the proposed approach. I would recommend the authors to look at the excellent continental water balance re-analysis which has been produced by the European Earth2Observe project : https://wci.earth2observe.eu/portal/. As this work is performed with multiple models and at 1/4◦ degree resolution, there is some interesting information to be drawn for this study of the Balearic islands.*

We agree with the reviewer that the water balance in terms of P-E should be more strongly justified in our case.

We have considered the data provided at the web site https://wci.earth2observe.eu/portal/ for the three grid points (resolution 0.25º) closest to the airports of Mallorca, Menorca and Ibiza. Monthly total precipitation values (PCP) for each grid point have been extracted from 1981 to 2010 and the mean monthly values have been computed (these monthly data originally come from the ECMWF data set). Monthly total values of evapotranspiration (EVT; i.e. surface evaporation, interception and transpiration) and monthly total runoff (R; i.e. surface runoff, sub-surface flow and deep percolation) provided by the 8 available models have also been obtained from the Earth2Observe web site. For each model and variable, the mean monthly values with reference to 1981-2010 have been calculate. Finally, the 8 models-ensemble mean and inter-model standard deviation of the previous monthly values were obtained. With these values the water balance was estimated at each of the three mesh points considered. This balance is built as the precipitation minus the actual evapotranspiration minus the losses (WB = PCP-EVT-R)

These results reveal:

a) For the mesh point near the airport of Mallorca the precipitation values used by the models are much higher than those observed at the airport. As an example, the observed mean annual value (1981-2010) is 411.3 mm, while the same rainfall product used by the models is 597.4 mm. Regarding EVT, the model ensemble mean values are higher than those obtained at the Mallorca airport using the Thornthwaite formula, especially in summer. The monthly standard deviations are very high (that is, large differences among the different models). The potential evapotranspiration (PET) for the airport of Mallorca lies within the ensemble spread region. Regarding the water balance, and accepting 100 mm as saturation threshold for the soil, saturation in the Earth2Observe data is obtained during December, January and February (in our results there is no saturation at the airport) that may be due to the high precipitation values used in the models. Dryness is obtained in July and August and very low reserve values in June and September. In our results a remarkable water deficit is obtained in September (Fig 6), because the temperatures are still high. These results are displayed in the attached figures (components of the water balance and the balance itself).

b) For the grid point near Menorca airport, the monthly values of precipitation used by the models are much more similar to those observed at the airport (registered annual average of 548.6 mm versus 601.2 mm in the models). The EVT shows a behavior similar to that at the grid point near the airport of Mallorca: values are greater than those of PET obtained from the airport of Menorca data using the Thornthwaite expression, and there is a large spread among the 8 models. The calculated PET values are also well encompassed by the ensemble dispersion band. Regarding the water balance, saturation of the soil is obtained in January and February and a value close to saturation in December. Dryness is also obtained in July and August. These results are very similar to those obtained directly in our study for the Menorca airport.

c)      For the grid point close to the Ibiza airport, the average monthly precipitation values used by the models are also significantly higher than those registered at the Ibiza airport (observed annual mean of 411.1 mm versus 497.2 mm in the models). Again the average EVT values of the ensemble are larger than PET values given by the Thornthwaite's expression at Ibiza airport. The inter-model spread is very high. Regarding the balance, saturation of the soil

is not reached in any month; in contrast dryness is present during May, June, July and August. These results are in agreement with the results obtained directly with the airport data.

In conclusion, it seems that the method used in the paper is sufficient to obtain a clear representation of the drought object of the study.

Note we do not include the figures corresponding to Menorca and Ibiza analyses as they are very similar to the included figure.

A discussion of the results based on the Earth2Observe models and tools (including a figure) and its comparison with those obtained directly with the airport data, will be added in the new version of the paper.

[Figure]

[Figure]

*d) The hydrological year 2015-16 is characterized by very intense rainfall events in September and then a continuous deficit for the rest of the rainy period. This begs the question of the role of runoff. Especially when the season starts with heavy precipitation, on soil dried out by the summer, much less water will infiltrate and thus recharge soil moisture. An analysis of the Earth2Observe multi-model ensemble or river discharge data could offer some more insight into the other terms of the water balance equation. It would shows how far the simple P-E balance can be trusted. Furthermore the authors use PET and E, so there is a further assumption to be tested.*

We understand your comments on the runoff issue. No data is collected in streams about any possible discharge occurred in September 2015. The very few measurements belong only to special cases and always before 2014. Note that in the Balearics there are not permanent rivers. In addition, no information about the episode can be obtained from the Earth2Observe web page, since model data extends only till 2012.

However, the runoff coefficient for a nearby river basin to the Mallorca airport (few kilometers away) was estimated by García et al. (2017), based on daily observed stream flow series for the 1977-2009 period. The estimated runoff coefficient was as low as 0.03. This result ensures that the conversion of precipitation into surface runoff is quite low for this nearby basin. Furthermore, no substantial changes are found in the spatial distribution of the physiography

and hydrology of the stream basin where the meteorological station is located. We can safely assume that almost of precipitation is infiltrated and that the P-E balance is quite realistic when assessing the climatic water balance for the 1981-2010 period. Even for the heavy precipitation event at the end of the 2015 warm season, most of precipitation should infiltrate owing to the high infiltration capacity of the soil and its low water content.

García, C., Amengual, A., Homar, V., and Zamora, A. (2017). Losing water in temporary streams on a Mediterranean island: Effects of climate and land-cover changes. Global and Planetary Change, 148, 139-152.

*Minor Comments :*

*1.I guess it is OK to reference general media in a scientific publication in order to highlight the societal relevance of the topic studied. But this should not exempt the authors from putting proper references. So please cite the newspaper which served as a source of the information.*

We agree. We will cite the newspaper (Diario de Mallorca).

*2. 3rd paragraph of Introduction : "somehow characterizes the type of natural vegetation". This seems very vague especially when you consider that there are vegetation classification which could be used to refine such a statement.*

The phrase will be modified eliminating the vagueness. We will do specific reference to the vegetation types that live in the Balearics.

*3. Is it necessary to repeat the inter-annual analysis for the civil and hydrological years (figures 3 and 5) ? As pointed out above the spatial representativeness seems more critical to me.*

Figure 3 will be eliminated

*4. It is unclear for which year figure 6 is ! This is neither explained in the text nor the caption of the figure.*

Figure 6 does not correspond to a specific year. It presents the climatic water balance at the three airports. This is indicated in the caption.

*5. Many figures do not have units on the axis or title and we have to retrieve that information from the caption. In general the figures are of poor quality.*

We will modify the figures to produce better quality versions.

*6. Description of figure 8 in the text and its caption do not correspond. Are three stations used or only the airport data ?*

The figure shows the time series of the accumulated precipitation from November to January only for the Mallorca airport, as it is indicated in the caption. We will explain this it in the text.

**7. End of conclusion : The natural vegetation in the Balearic islands is not wild I think !!**

OK, we will improve the sense of the sentence.

---

## Author Comment (AC2) · 26 Sep 2017

**Reply to the comments of referee 2 (Dr Guijarro)**

Thanks for your kind comments that, undoubtedly, will help to improve the first version of the manuscript.

*General comments: I agree with referee #1 on the importance of the subject and the convenience of the publication of this work.*

Thanks for your good opinion.

*It is also clear the need to use empirical formulas to calculate evapo-transpiration rates, due to the lack of direct measurements. In this sense, the Hargreaves method is much used when the variables observed at a site are limited. However, the airports are observatories with a more complete range of observations, and therefore more complex alternatives, such as the Penman-Monteith method, could have been used. Unfortunately, radiation is measured only at the Palma airport and, while a comparison on ET values computed with Penman-Monteith and Hargreaves formulas would assess the validity of the latter within the studied climatic area, that effort could give birth to a new article by itself.*

We have used the Hargreaves formula to calculate the daily potential evapotranspiration (PET) during the hydrological year 2015-16 because the data provided by the three airports of the Balearic Islands, and by the used automatic weather stations, are appropriate for this methodology. The Penman-Monteith method would probably provide more accurate values of PET, but as pointed out by the reviewer, this approach is not applicable in our study owing to the lack of the required data. We agree that a comparison between the PET values provided by the two methods at the Mallorca airport would be very interesting, not only for the hydrological year analyzed in our study but also from a climatologic point of view.

*More debatable is, as referee #1 also points out, the use of both Thornthwaite monthly hydrological balances and daily balances with the Hargreaves estimates. Thornthwaite approach has been extensively used in the past when dealing with monthly series, and can still be used for comparison purposes, especially on monthly climatic averages. However, it has been reported that Thornthwaite method under-estimates ETP in arid climates, but if more realistic empirical formulas are used (Guijarro, 1986, cited by the authors), monthly hydrological balances following Thornthwaite result in soils being completely dry all year round in extensive zones of the Balearic islands, which is also unrealistic. Therefore, climatic balances should be derived by applying daily balances to the reference period and then computing the monthly values, or at least the Thornthwaite method should also be applied to the 2015-16 year to assess the impact of using these different approaches.*

The Thornthwaite approach applied for obtaining PET monthly climatic values uses directly the monthly mean temperatures provided by AEMET for the period 1981-2010. The aim is to build a reference water balance for a comparison with the particular water balance of the hydrological year 2015-16. For this hydrological year, the PET monthly values have been calculated from the daily values obtained by the Hargreaves method. We agree that some comparison between the two methods for this year is necessary to fully justify the reference to

the climatic water balance. Although not included in the paper, a comparison between both methods was done in the course of the study. Specifically, monthly PET values using Thornthwaite method were calculated for the hydrological years 2014-15 and 2015-16 at the E1 site in Mallorca. Analogous monthly values were obtained from the daily PET values given by the Hargreaves formula.  The E1 station is located in the most arid region of the island. The two time series show a correlation coefficient of 0.9 (see Figure). For the warmer/colder months the Thornthwaite method reveals larger/lower monthly PET values than the other approach. This discussion will be included in revised version of the paper.

[Figure]

*Specific comments:*

*115: "These are the longest homogeneous climatic series without gaps..." Has their homogeneity been assessed? I would remove ' homogeneous' otherwise.*

We agree. We will remove 'homogeneous'.

*With respect to comments on lines 132-136 and 155-156 a reference could be added to the sentence in line 156: "do not respond to the same circulation patterns, as previously reported by Guijarro (2002 and 2003)" GUIJARRO JA (2002): Tendencias de la precipitación en el litoral mediterráneo español. In Guijarro et al. (Eds.), El agua y el clima, Asociación Española de*

*Climatología, A-3:237-246, ISBN 84-7632-757-9. GUIJARRO JA (2003): El flujo geostrófico superficial en el Mediterráneo Balear durante el periodo 1948-2002. Rev. climatol., 3:45-59.*

References will be included

*The statement in lines 137-138 is debatable. Different regimes can be seen in different parts of Mallorca (Sumner et al., 1995). SUMNER G., GUIJARRO J.A., RAMIS C. (1995): The impact of surface circulation on significant daily rainfall patterns over Mallorca. International Journal of Climatology, 15:673-696.*

The statement will be revised. Some ideas from the indicated paper will be included.

*Technical corrections:*

*131: 'end'→'and'*

This will be corrected.

*254: "to which the local vegetation was subjected to." Too many 'to's? Remove the last one?*

Sentence will be changed as suggested.

---

## Referee Report (RR1)

**Review of "On The Drought In The Balearic Islands During The Hydrological Year 2015-2016" by Ramis et al.**

This paper studies the extreme droughts of 2015-16, or more generally on the water resources, of the Balearic islands covers a rather important topic and after revision offers a solid analysis. Compared to the first version, the study now includes a strong argumentation to demonstrate that their simple methods they use hold their rank and are suitable for the proposed interpretation.

I appreciated that the comparison with state of the art land surface models was performed. It is quite sobering to see that they still cannot reach the resolutions needed to address the water cycle of small islands. I also liked that now the authors verify that changing their methodology for estimating potential evaporation has no consequence on the interpretation. The representativeness of the rainfall measurements used is now also discussed. Thus, I would like to recommend to accept this paper perhaps a few minor changes.

As usual, there are some minor things which can be improved. I would like to propose those which I noted while reading and for the benefit of the authors.
- "according to media" could be replaced by a proper reference ! El Pais or Die Inselzeitung (:-)) should be acceptable references for NHESS, or ?
- Lines 141-143 : There are a number of studies on the spatio/temporal variability of rainfall and some proposed metrics. It would be nice to have a reference here so that interested readers can go further.
- Lines 200-217 : The authors might consider to present their water balance model with a simple equation. For some readers it will be simpler to understand than a verbose paragraph.
- Line 250 : The 3 hourly rainfall which has driven the land surface models in the Earth2Observe simulations is not coming from the ECMWF re-analysis but rather independent satellite estimates : Beck et al. MSWEP: 3-hourly 0.25∘ global gridded precipitation (1979–2015) by merging gauge, satellite, and reanalysis data, Hydrol. Earth Syst. Sci., 21, 589–615, 2017. It would be good to add this reference so that we know the origin of the error you identify.
- The high precipitation values used in the Earth2Observe simulations certainly comes from spatial issues in MSWEP. It cannot properly make the difference between the valley in Mallorca and the mountains to the West.
- Line 346 : "really dry" not the right wording in a scientific paper, or ? "extremely dry" is perhaps more suitable.
- Line 373 : "continuous deficit" does not sound right to me but I see what you mean.
- Line 412 : "essentially unfavorable" is the word essentially needed ?
- Line 464 : "... was also unappreciable over all three islands." sounds better to me.

---

## Author Response (AR2)

**Reply to the comments of reviewer 1 to the second version of the paper.**

Thanks for your comments and your opinion about the paper.

All the suggestions have been incorporated to the paper except that we consider that the inclusion of an equation for the water balance is not necessary. The explanation of the method we provide in the text is enough for its understanding.

**Reply to the comments of reviewer 2 to the second version of the paper.**

Thanks for your opinion.